# Local guidelines for admission to UK midwifery units compared with national guidance: A national survey using the UK Midwifery Study System (UKMidSS)

**Ceri Glenister[1], Ethel Burns[1], Rachel Rowe[2]\***

**1** Oxford School of Nursing and Midwifery, Faculty of Health and Life Sciences, Oxford Brookes University, Oxford, United Kingdom, **2** Nuffield Department of Population Health, NIHR Policy Research Unit in Maternal and Neonatal Health and Care, National Perinatal Epidemiology Unit, University of Oxford, Oxford, United Kingdom

\* rachel.rowe@npeu.ox.ac.uk

**Data Availability Statement:** Data cannot be shared publicly because to do so could potentially identify individual maternity units or services, and assurances were given to respondents that any

## Abstract

### Objectives

To describe the extent to which local guidelines for admission to UK midwifery units align with national guidance; to describe variation in individual admission criteria; and to describe the extent to which alongside midwifery units (AMUs) are the default option for eligible women.

### Design

National cross-sectional survey.

### Setting

All 122 UK maternity services with midwifery units, between October 2018 and February 2019.

### Outcome measures

Alignment of local admission guidelines with national guidance (NICE CG190); frequency and nature of variation in individual admission criteria; percentage of services with AMU as default birth setting for eligible women.

### Results

Admission guidelines were received from 87 maternity services (71%), representing 153 units, and we analysed 85 individual guideline documents. Overall, 92% of local admission guidelines varied from national guidance; 76% contained both some admission criteria that were 'more inclusive' and some that were 'more restrictive' than national guidance. The most common 'more inclusive' admission criteria, occurring in 40–80% of guidelines, were: explicit admission of women with parity $\geq$4; aged 35-40yrs; with a BMI 30-35kg/m$^2$;

publication would not do so. Requests for access to the data underlying our findings will be considered by the National Perinatal Epidemiology Unit Data Sharing Committee and should be addressed to jenny.kurinczuk@npeu.ox.ac.uk.

**Funding:** This research is funded by the National Institute for Health Research (NIHR) Policy Research Programme, conducted through the Policy Research Unit in Maternal and Neonatal Health and Care, PR-PRU-1217-21202 (RR). The views expressed are those of the authors and not necessarily those of the NIHR or the Department of Health and Social Care. The funders had no role in study design, data collection and analysis, decision to publish, or preparation of the manuscript.

**Competing interests:** The authors have declared that no competing interests exist.

selective admission of women with a BMI 35-40kg/m$^2$; Group B Streptococcus carriers; and those undergoing induction of labour. The most common 'more restrictive' admission criteria, occurring in around 30% of guidelines, excluded women who: declined blood products; had experienced female genital cutting; were aged <16yrs; or had not attended for regular antenatal care. Over half of services (59%) reported the AMU as the default option for healthy women with straightforward pregnancies.

## Conclusions

The variation in local midwifery unit admission criteria found in this study represents a potentially confusing and inequitable basis for women making choices about planned place of birth. A review of national guidance may be indicated and where a lack of relevant evidence underlies variation in admission criteria, further research by planned place of birth is required.

## Introduction

Since the early 1990s United Kingdom (UK) maternity care policy has supported women's choice of planned place of birth and increased access to midwifery-led models of intrapartum care for healthy women with straightforward pregnancies, and this is now supported by maternity care strategy and guidance documents in all four countries of the UK [1–8]. There is robust evidence about the safety of midwifery-led settings for these women in terms of lower chances of intrapartum interventions and comparable maternal and neonatal outcomes to obstetric settings [9–14], and about benefits in terms of women's satisfaction with their birth experience [10–13]. Psychological safety through access to choice and retaining a sense of control in childbirth is important to women [15], as is the chance of having a straightforward birth [16], and a positive childbirth experience [17]. In the UK, where midwifery-led settings are an integrated part of NHS maternity care (Health and Social Care in Northern Ireland), this evidence is recognised in national guidance and strategy documents which recommends that four birth place options: obstetric unit (OU), home and two types of midwifery unit (MU), are offered by providers of maternity services [3,5,6,8].

Planned birth in an MU is considered "particularly suitable" for healthy women with straightforward pregnancies [3]. An MU is "a location offering maternity care to healthy women with straightforward pregnancies in which midwives take primary professional responsibility for care" [18,19] and can be either 'alongside' (AMU), located on the same site as a consultant-led OU, or 'freestanding' (FMU), located on a separate site, away from an OU. The number of MUs has increased across all four countries of the UK, most notably in the case of AMUs in England which increased from 26 to 132 between 2007 and 2019 [20–22]. In England, the number of FMUs has remained fairly static over the same period and remains fewer than AMUs [22,23], but the picture in the rest of the UK is varied. In Scotland in 2017 there were six AMUs and 19 FMUs [5]; in Northern Ireland the first AMU opened in 2001 and by 2014 there were six AMUs and three FMUs [24]; in Wales in 2018 there were eight AMUs and 14 FMUs [25]. In 2015, in England, around 14% of women gave birth in an MU and it is estimated that over a third of women may be potentially eligible to do so [21].

In order to exercise choice over where to give birth, women and their midwives need accurate, unbiased information about the relative benefits and risks of the options available [4,26].

Discussion or assessment of risk with women is complex, and how best to identify whether a women is at 'higher' or 'lower' risk of complications has been the subject of debate and discussion [27–32], but the aim of risk assessment is to screen for where an intervention could improve outcomes [29,33]. One of the complications of assessment criteria for place of birth is that the 'intervention' has changed from being admission to hospital, to 'permission' to plan birth outside of an OU [34], which may have led to MUs developing their own admission criteria independently of national guidance [33]. In the UK there is clear national guidance to support decision making about planned place of birth [3], but there is some evidence that admission criteria used by MUs may vary between units and depart from national recommendations [33,35,36], and it is unclear whether MUs or OUs are the default option for women with straightforward pregnancies. This variation may result in inequality of access to MUs for women, inconsistent application of evidence about the relative benefits and risks, and potential confusion about which women may be most suitable for MU care.

We aimed to document and describe variation in local maternity service guidelines for planned admission for intrapartum care ('admission guidelines') to AMUs and FMUs across the UK. Our main objectives were to describe the extent to which admission guidelines aligned with or varied from national guidance; explore whether this variation was associated with selected characteristics of maternity services; explore and describe variation in individual MU admission criteria; and describe admission policies in AMUs, i.e. whether the AMU was the default for women considered suitable (opt-out) or whether women needed to actively request birth in the AMU (opt-in).

## Methods

### Study design

We carried out a national cross-sectional survey to collect and describe MU admission guidelines. National guidance in the form of the NICE guideline CG190: Intrapartum care for healthy women and babies [3] was chosen as a reference against which to measure variation. While this guideline applies formally in England and Wales, guidance in Scotland and Northern Ireland is also broadly in line with NICE CG190.

### Data and sources

The sampling frame for this study was all 122 maternity services (NHS Trusts and Health Boards) in the UK with at least one MU. We used the UK Midwifery Study System (UKMidSS), a national infrastructure for carrying out research in MUs across the UK, for this survey [37]. We emailed UKMidSS reporters in all maternity services with AMUs, and midwives nominated by Heads of Midwifery in services with FMUs only, in October 2018, introducing the study and giving a link to a brief study-specific online survey. As part of this survey we requested their current guidelines for admission to MUs, with a request to send separate AMU and FMU guidelines if these were in use. Up to six reminder emails were sent to non-responders and the survey was closed in February 2019.

In the survey we collected data about the number of AMUs and FMUs in each maternity service; the number of years each unit had been open for; AMU admission policy ('opt-out', i.e. AMU default option for eligible women; or 'opt-in', i.e. women required to actively chose AMU); and whether AMUs and FMUs in the service used the same admission guideline.

Data about the number of births in each maternity service were obtained from the MBRRACE-UK Perinatal Mortality Surveillance Report for 2016 [38]. Data about the number of births in each AMU came from the UKMidSS Severe Obesity Study in 2016 [39].

### Admission guideline handling and data extraction

On receipt, admission guidelines were assigned an identification number to enable matching to corresponding survey response and additional births data. Some admission guidelines received were used by more than one maternity service; these were counted only once in the guideline analyses, but all maternity services covered by that guideline were considered to have returned a guideline for the purposes of calculating response. Some maternity services with more than one AMU/FMU, in which different units used separate admission criteria, sent more than one guideline. These were reviewed as separate guidelines in the guideline analyses. Some maternity services sent a single guideline which contained separate criteria for admission to AMUs and FMUs. These were reviewed as separate AMU and FMU guidelines.

Each guideline was read in full by CG prior to data extraction. Data about guideline characteristics and content were systematically extracted by CG and entered into a custom designed data collection tool, created to be responsive to guideline variation. Duplicate data extraction was not carried out because of resource constraints, but CG discussed any uncertainties with RR during data extraction.

**Guideline characteristics.** The study drew on the AGREE instrument for systematically evaluating guideline quality [40], to capture some characteristics that might be indicative of guideline quality, e.g. date of guideline, guideline authorship and evidence base. Other guideline characteristics extracted were: guideline length (number of pages), number of admission criteria, and whether there was an explicit care pathway for women wishing to give birth outside of guideline recommendations.

**Admission criteria.** Four tables from NICE Guideline CG190 [3], listing specific criteria to be assessed for women planning place of birth, were used as a reference against which to compare the individual admission criteria listed in each guideline (Tables in S1 File). CG compared each criterion listed in each guideline to the information in these tables and categorised them in one of the following ways:

'More restrictive' than NICE CG190:

- Criteria not listed in NICE CG190, for which the MU required women to have an individual assessment for admission

- Criteria for which NICE CG190 recommends individual assessment, but the MU did not admit women with the specific criteria, irrespective of individual assessment

- Criteria not listed in NICE CG190, for which the MU did not admit women with the specific criteria.

 'More inclusive' than NICE CG190:

- Criteria for which NICE CG190 recommends OU birth, but where the MU explicitly admitted women with the specific criteria

- Criteria for which NICE CG190 recommends OU birth, but where the MU offered admission following individual assessment

- Criteria for which NICE CG190 recommends individual assessment, but where the MU explicitly admitted women without requiring individual assessment.

Individual criteria that were 'more inclusive' were only considered as such if they were explicitly listed in admission guidelines; no inferences were made about MUs admitting women if criteria from Tables 6–9 of NICE CG190 were not listed.

For each guideline, the individual admission criteria that were 'more restrictive' or 'more inclusive' than those listed in NICE CG190 were extracted by CG. For a small number of specific commonly-occurring admission criteria that were rarely aligned with NICE CG190, where there was substantial variation between guidelines, CG extracted more detailed information about each admission criteria and how they varied.

Using these categorisations of admission criteria as 'more restrictive' or 'more inclusive' CG categorised each guideline in one of the following ways:

- Aligned with NICE CG190: guidelines which explicitly stated that the admission criteria used were those listed in NICE CG190 or which reproduced the tables from CG190 in the guideline

- 'More restrictive' than NICE CG190: guidelines in which at least one admission criterion was 'more restrictive', and no criteria were 'more inclusive'

- 'More inclusive' than NICE CG190: guidelines in which at least one criterion was 'more inclusive' and no criteria were 'more restrictive'

- 'Both more restrictive and more inclusive' than NICE CG190: guidelines in which at least one criterion was 'more restrictive' and at least one criterion was 'more inclusive'.

## Analysis

All data extracted from the guidelines were imported into Stata 15 statistical analysis software [41] and merged with data from other sources (data about number of births from MBRRA-CE-UK and UKMidSS) to produce a single dataset. We summarised the data generating descriptive statistics as frequencies and percentages. We tested for the presence of response bias by using the Chi-square test to compare selected characteristics of maternity services (AMU/FMU configuration, number of births per year, percentage of AMU births) that did and did not send an admission guideline. We also used the Chi-square test to explore associations between selected guideline and maternity service characteristics and the extent to which guidelines were aligned with national guidance.

## Patient and public involvement

Lay members of the Co-investigator Group for the NIHR Policy Research Unit in Maternal and Neonatal Health and Care, and the UKMidSS Steering Group, were involved in discussing the research questions for this study, interpretation, and will be involved in further dissemination of the results.

## Ethics statement

Using the Health Research Authority classification tool for research for England [42], this study was classified as research not requiring NHS research ethics approval. Information about the aims of the study and how the results would be used was included in the invitation email, and return of a completed survey response and/or guideline was considered as consent to take part.

## Results

### Response rate and configuration of care

Overall, 122 maternity services across the UK were identified as having at least one MU, with 216 MUs in total (Fig 1, S1 Table). Complete survey responses were received from 102 services

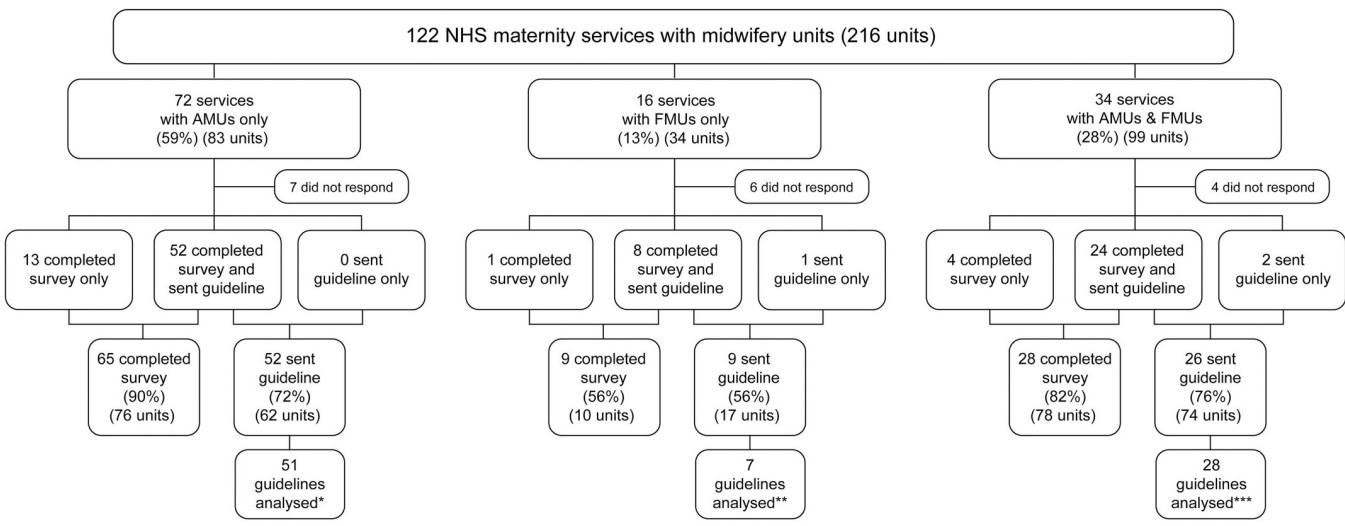

* Two services used the same guideline
** Two guidelines received did not include individual admission criteria
*** Two guidelines analysed as separate AMU & FMU guidelines (i.e. as four separate guidelines)

**Fig 1. Number of maternity services completing survey and sending a guideline, and number of guidelines analysed, by configuration of care.**

(84%), representing 164 MUs, and guidelines were received from 87 services (71%) representing 153 MUs. All maternity services in Northern Ireland reported using the GAIN guideline [24], which was reviewed once. Most MUs in Wales reported using the All Wales Midwife-led Care Guidelines [43], which was also reviewed once. In total, 85 separate guidelines were included in the analyses.

There was no statistically significant difference between the characteristics of maternity services that did and did not send a guideline (Table 1 in S2 File). Services with FMUs only were less likely to complete the survey ($p = 0.004$, Table 2 in S2 File). This difference was also reflected in the finding that services with <10% of AMU births per year were less likely to respond to the survey.

## AMU admission policy (opt-in vs opt-out)

Over half (59%) of maternity services with AMUs reported operating an opt-out policy whereby the AMU was the default planned place of birth for eligible women (S2 Table). AMU admission policy was not statistically associated with configuration of care, the length of time the longest standing AMU in the service had been open, the number of births per year in the service, or the percentage of AMU births in the service, but services where the AMU had an opt-out policy had higher proportions of births in the AMU, compared with those where the AMU had an opt-in policy.

## Guideline characteristics

The characteristics of the guidelines are presented in Table 1. Most guidelines (71%) gave a publication date and 57 of these (95%) had been written or updated in the previous three years (January 2015 to December 2018). Almost a third of guidelines (31%) did not list the author(s) or a guideline development group. Of those that did, there were three guidelines in which there was evidence of service user involvement in the development of the guideline (5% of those for whom authors were listed and 3.5% of guidelines analysed). Guidelines ranged from 1–100 pages in length, with around a quarter of guidelines (26%) comprising four pages or

**Table 1. Characteristics of guidelines.**

| Characteristic | Number of guidelines | |
|---|---|---|
| | **n** | **%** |
| **Authorship** | | |
| None listed | 26 | 30.6 |
| Author listed | 50 | 58.8 |
| Guideline development group | 9 | 10.6 |
| **Service user involvement** | | |
| No | 82 | 96.5 |
| Yes | 3 | 3.5 |
| **Length (number of pages)** | | |
| 1–4 | 22 | 25.9 |
| 5–14 | 23 | 27.1 |
| 15–24 | 22 | 25.9 |
| 25–103 | 18 | 21.2 |
| **Evidence base** | | |
| None listed | 34 | 40.0 |
| Reference list | 48 | 56.5 |
| Evidence reviewed | 3 | 3.5 |
| **Number of admission criteria** | | |
| 8–31 | 22 | 25.9 |
| 32–64 | 21 | 24.7 |
| 65–88 | 21 | 24.7 |
| 89–153 | 21 | 24.7 |
| **Outside guideline referral pathway** | | |
| No | 40 | 47.1 |
| Yes | 45 | 52.9 |

less. This was partly attributable to the fact that some MUs sent only the guideline pages listing admission criteria and some sent entire guidelines. A quarter of guidelines (25%) listed more individual admission criteria than the 88 listed in Tables 6–9 in NICE CG190. Just over half of the guidelines (57%) cited at least one reference as an evidence base for their recommendations. These ranged from a single reference to NICE CG190, to a page of references to recent research articles. Three guidelines explicitly discussed the evidence base behind the recommendations. Just over half of guidelines (53%) specified a referral pathway for women who wished to birth outside of an OU, but fell outside of the MU eligibility criteria.

## Admission guidelines compared with national guidance

The extent to which admission guidelines were aligned with NICE CG190 is presented in Table 2. Overall, over three quarters of guidelines (77%) listed both some criteria that were 'more restrictive' than those listed in NICE CG190 and other criteria that were 'more inclusive'. Less than one in ten of the guidelines (7%) were aligned explicitly with the criteria listed in NICE CG190, although a further 18 guidelines (21%) had a small number of variant criteria (1–5) and some of these were otherwise similar in layout and content to NICE CG190. The number of individual criteria in each guideline that varied from NICE CG190 ranged from 1–19.

None of the guideline or maternity service characteristics studied were statistically significantly associated with the extent to which guidelines aligned with national guidance, but numbers of guidelines in some groups were very small.

**Table 2. Guideline alignment with NICE CG190 and selected characteristics of guidelines and maternity services.**

| | Aligned with NICE (n = 6) | | Either more restrictive or more inclusive (n = 14) | | Both more restrictive and more inclusive (n = 65) | | All guidelines (n = 85) | | p-value |
|---|---|---|---|---|---|---|---|---|---|
| | n | % | n | % | n | % | n | % | |
| **Scope of guideline** | | | | | | | | | |
| AMU only | 4 | 66.7 | 9 | 64.3 | 46 | 70.8 | 59 | 69.4 | |
| FMU only | 1 | 16.7 | 2 | 14.3 | 10 | 15.4 | 13 | 15.3 | |
| AMU & FMU | 1 | 16.7 | 3 | 21.4 | 9 | 13.9 | 13 | 15.3 | 0.97 |
| **Number of criteria** | | | | | | | | | |
| 8–31 | 0 | 0.0 | 4 | 28.6 | 15 | 24.6 | 22 | 25.9 | |
| 32–64 | 0 | 0.0 | 2 | 14.3 | 19 | 31.2 | 21 | 24.7 | |
| 65–88 | 6 | 100 | 3 | 21.4 | 11 | 18.0 | 21 | 24.7 | |
| 89–153 | 0 | 0.0 | 5 | 35.7 | 16 | 26.2 | 21 | 24.7 | 0.645[a] |
| **Number of births per year**[b] | | | | | | | | | |
| <3,500 | 2 | 33.3 | 5 | 35.7 | 13 | 20.0 | 20 | 23.5 | |
| 3,500–4,999 | 1 | 16.7 | 3 | 21.4 | 15 | 23.1 | 19 | 22.3 | |
| 5,000–5,999 | 0 | 0.0 | 4 | 28.6 | 16 | 24.6 | 20 | 23.5 | |
| 6,000–17,000 | 3 | 50.0 | 2 | 14.3 | 21 | 32.3 | 26 | 30.6 | 0.519 |
| **% of births in AMU** | | | | | | | | | |
| <10 | 1 | 20.0 | 2 | 14.3 | 7 | 11.3 | 10 | 12.4 | |
| 10.1–15 | 2 | 40.0 | 5 | 35.7 | 17 | 27.4 | 24 | 29.6 | |
| 15.1–20 | 1 | 20.0 | 4 | 28.6 | 24 | 38.7 | 29 | 35.8 | |
| 20.1–39 | 1 | 20.0 | 3 | 21.4 | 14 | 22.6 | 18 | 22.2 | 0.959 |

[a] Chi-squared test excluding the 6 guidelines that aligned with NICE CG190 as they all listed 88 admission criteria.
[b] Overall annual number of births in the maternity service (NHS Trust or Health Board).

As in NICE CG190, all guidelines specified that current pregnancies should be singleton, cephalic and >37 weeks' gestation, with most guidelines specifying an upper gestational limit of either 41$^{+6}$ or 42$^{+0}$ weeks. All guidelines listed some specific admission criteria, with no guideline giving only non-specific criteria such as 'straightforward pregnancy' or 'suitable for midwifery-led care'.

## Individual admission criteria compared with national guidance

Overall, 73 guidelines (86%) listed at least one admission criterion which was 'more inclusive' than NICE CG190 and 71 guidelines (84%) listed at least one admission criterion which was more restrictive than NICE CG190. In total, 53 individual admission criteria were identified that varied from national guidance, 26 that were 'more inclusive' (Tables in S3 File) and 27 that were 'more restrictive' (Tables in S4 File). The most frequently occurring individual 'more inclusive' and 'more restrictive' criteria across all guidelines are shown in Table 3.

Parity was the most frequently occurring 'more inclusive' criterion, but also had varying limits which are listed in more detail in Table 4. Maternal age 35–40 years and women with a BMI 30-35kg/m$^2$, are criteria for which NICE CG190 recommends women are individually assessed, but in 54 (64%) and 47 (55%) admission guidelines respectively, these women were eligible for admission without an individual assessment. Women with a BMI 35-40kg/m$^2$ (for whom NICE CG190 recommends birth in an OU) were listed either for admission or for individual assessment in 37 guidelines (44%), with multiparous women in this group specified in

**Table 3. Most frequently occurring individual 'more inclusive' and 'more restrictive' criteria across all guidelines.**

| | Guidelines in which criteria were listed | |
|---|---|---|
| | n | %[a] |
| **'More inclusive' criteria[b]** | | |
| Parity ≥4 | 57 | 67.1 |
| Maternal age 35–40 years | 54 | 63.5 |
| BMI 30-35kg/m$^2$ | 47 | 55.3 |
| BMI 35-40kg/m$^2$ multiparous or any parity | 37 | 43.5 |
| Group B Streptococcus colonisation | 37 | 43.5 |
| Induction of labour, one intervention | 34 | 40.0 |
| **'More restrictive' criteria[c]** | | |
| Declining blood products | 26 | 30.6 |
| Female genital cutting | 24 | 28.2 |
| Maternal age <16years | 23 | 27.1 |
| Late booking/no antenatal care | 23 | 27.1 |

[a] Percentage of all guidelines (n = 85).

[b] Women explicitly eligible for MU intrapartum care or considered for admission following individual assessment.

[c] Women not mentioned in NICE CG190 explicitly excluded from or individually assessed for MU care.

21 guidelines. Group B Streptococcus colonisation and induction of labour requiring one intervention, for both of which NICE CG190 recommends OU birth, were listed as a reason to individually assess or to admit women in 37 (44%) and 34 guidelines (40%) respectively.

None of the most frequently occurring 'more restrictive' criteria (Table 3) were listed in NICE CG190. Declining blood products and having experienced female genital cutting were listed as a reason to individually assess or not to admit women in almost a third of all guidelines (31% and 28% respectively). Both maternal age less than 16 years and inadequate antenatal care were listed in over a quarter of guidelines overall (27%).

For admission criteria in relation to parity, previous PPH, maternal anaemia and prolonged rupture of membranes there was widespread variation from NICE CG190, the extent of which is shown in Table 4. Parity was specified as an admission criterion in 70 guidelines (82%), and less than a fifth of these (19%) were aligned with NICE CG190. A limit on previous postpartum blood loss was listed in 69 guidelines (81%), with 41% of these aligned with NICE CG190. A specific limit on maternal anaemia (Hb level in g/L) was mentioned in 78 guidelines (92%), a quarter of which (25%) were aligned with NICE CG190.

## Discussion

### Key findings

Over half (59%) of maternity services reported that their AMU was the default option for healthy women with straightforward pregnancies.

Admission guidelines varied considerably in their layout and characteristics, most notably in the number of admission criteria listed, and very few included an evidence base for the recommendations.

Few guidelines (<8%) were fully aligned with national guidance and 53 separate admission criteria were identified that departed from national recommendations. Over three-quarters of guidelines contained both criteria that were more inclusive and criteria that were more restrictive than national guidance.

**Table 4. Detailed midwifery unit admission criteria for parity, previous PPH and maternal anaemia.**

| Admission criteria | Number of guidelines | |
|---|---|---|
| | n | %[a] |
| **Parity** | | |
| ≤4 | 19 | 22.4 |
| ≤5 | 28 | 32.9 |
| ≤6 | 9 | 10.6 |
| No limit | 1 | 1.2 |
| Same as NICE CG190[b] | 13 | 15.3 |
| Not mentioned | 15 | 17.7 |
| **Previous PPH** | | |
| <0.5 litre | 5 | 5.9 |
| <1 litre | 23 | 27.1 |
| <1.5 litre | 3 | 3.5 |
| <2 litre | 1 | 1.2 |
| Subsequent normal blood loss | 1 | 1.2 |
| No previous PPH | 8 | 9.4 |
| Same as NICE CG190[c] | 28 | 32.9 |
| Not mentioned | 16 | 18.8 |
| **Maternal anaemia** | | |
| Hb≥85g/l | 18 | 21.2 |
| Hb≥90g/l | 20 | 23.5 |
| Hb≥95g/l | 6 | 7.1 |
| Hb≥100g/l | 11 | 12.9 |
| Hb≥105g/l | 2 | 2.4 |
| Same as NICE CG190[d] | 21 | 24.7 |
| Not mentioned | 7 | 8.2 |

[a] Percentage of all guidelines (n = 85).

[b] NICE CG190 recommends individual assessment for women of parity ≥4.

[c] NICE CG190 recommends planned OU birth for women with previous 'primary postpartum haemorrhage requiring additional treatment or blood transfusion'.

[d] NICE CG190 recommends individual assessment for women with Hb 85-105g/l at onset of labour.

Admission criteria that were more inclusive than national guidance tended to occur more frequently across guidelines than those that were more restrictive. The most frequently occurring more inclusive criteria included admission of women with parity of ≥4, maternal age 35-40yrs, BMI 30-35kg/m$^2$, selective admission of women with a BMI 35-40kg/m$^2$, Group B streptococcus colonisation and selective induction of labour. The most frequently occurring more restrictive criteria excluded women declining blood products, having experienced female genital cutting, with maternal age <16yrs and inadequate antenatal care.

Parity, previous PPH and maternal anaemia were listed as admission criteria very frequently across guidelines, and varied widely, with most not in alignment with national guidance.

## Strengths and limitations

This is the first UK-wide study to document AMU admission policy (i.e. whether the AMU is the 'default' option for eligible women), to systematically document alignment with and variation from national guidance in local MU admission guidelines and in individual admission

criteria. Data were collected using UKMidSS [37], a well-established, effective national research infrastructure for MUs across the UK, and derived from other reliable sources. High response rates and a low probability of response bias increase the generalisability of our findings. Resource and time constraints meant that data extraction and analysis was carried out by one author only (CG), but any uncertainties were discussed with RR, and this is unlikely to have introduced systematic bias into our study. There was some evidence that maternity services with FMUs only were under-represented in our study, almost certainly because at the time the data were collected, UKMidSS had established midwife reporters in AMUs, but contact with FMUs was new. The number of maternity services with no AMUs and only FMUs is relatively small (13%), but nevertheless the findings of this study may be less generalisable to FMU admission guidelines. It should be noted, however, that current national (NICE) guidance does not recommend different admission criteria for AMUs and FMUs, although guidance used for the whole of Northern Ireland does do so [24]. No formal assessment was made of guideline quality and numbers for some comparisons were small. Finally, because of time constraints we surveyed maternity service providers at only one point in time so were unable to follow up to ask, for example, why their guidelines were not aligned with national guidance.

## Interpretation in the light of other evidence

Some limited data about assessment criteria for admission to midwifery-led settings and MUs have been collected in previous studies, all of which found variation in admission criteria [33,35,36]. The move away from non-specific criteria such as 'suitable for midwifery-led care' as observed by Campbell [33], towards specific admission criteria is in keeping with the current national context for evidence-based care [44]. However, the extent of variation found by our study, reveals a lack of consensus about how best to identify women who are likely to have a straightforward birth and are therefore suitable for planning birth in a MU. This lack of consensus may be historical [33], may indicate inconsistency in the application of the available evidence, or a lack of relevant evidence on which to base clinical recommendations [45], leading to local bias [46].

Variation from national guidance in specific admission criteria may also be indicative of issues of concern in local populations for which there is little research evidence to guide practice. Some of the 'more restrictive' admission criteria identified in our study (declining blood products, having experienced female genital cutting, maternal age >16yrs and inadequate antenatal care), may reflect this, but may also disproportionately affect women from religious and ethnic minorities, and those of lower socio-economic status. There is some evidence that women in these groups may be at a higher risk of adverse outcome. For example, women who are Jehovah's Witnesses have an increased risk of death and morbidity associated with obstetric haemorrhage [47,48]. There is some limited evidence that women who have experienced female genital cutting may have an increased risk an emergency Caesarean section and severe perineal trauma [49]. While young maternal age and fragmented or reduced antenatal care are both associated with adverse outcomes such as preterm birth and low birth weight, it is likely that the underlying causes of these adverse outcomes are socio-economic factors including social class, deprivation and smoking [50]. Although late presentation for antenatal care could lead to uncertain gestation and therefore place women outside of the criteria for MU care, none of the guidelines reviewed mentioned gestation in relation to late booking or fragmented antenatal care. None of the evidence about outcomes for women in these groups was considered sufficient justification to recommend OU birth during the development of NICE guidance [3].

AMU opt-out polices, whereby the AMU is the default option for eligible women, have the potential to increase equity of access to MU care. Our data suggest that maternity services with

an opt-out AMU may have a higher proportion of births in the AMU. It is important that any default birth place option is not implemented at the expense of informed choice of all the available options [4,26], particularly in the light of evidence of frequent closures of FMUs [51], as these are associated with optimal outcomes for healthy women and their babies [52]. It is also important that MU admission criteria, irrespective of opt-out or opt-in policy, do not disproportionately deny access to midwifery-led care for women from minority and socio-economically deprived backgrounds, yet some of the deviations from NICE guidance evidenced by our study may have this effect.

Our study has provided evidence that, in some specific areas, relevant research published since the development of national guidance may be driving more inclusive admission criteria, particularly with regard to the explicit inclusion of women aged 35–40 years, with a BMI in the range 30-35kg/m$^2$ and qualified inclusion of women with a BMI in the range 35-40kg/m$^2$ [39,53,54]. Research into alternative settings for induction of labour [55] may reflect women's desire for a different experience to that offered by conventional OU induction and the need for alternative management strategies in the context of rising induction rates [22,23], which may be driving the inclusion of these women in just under half of MU guidelines. The extent of widespread variation between admission guidelines for other specific criteria (multiparity, previous PPH and maternal anaemia) is evidence of further underlying uncertainty in relation to outcomes, perhaps particularly where there may be a continuum of increasing risk with no clear step-change. Ongoing studies into outcomes for women planning AMU birth after a previous PPH and the management and outcomes of women who experience a PPH in an MU, may help fill some evidence gaps [56].

Maternity services in England have a responsibility to enable NICE guidelines to be applied in their services, and providers in other UK countries should be guided by their own, similar, guidance or strategy documents. However, national clinical guidance is intended to guide, not prescribe, clinical practice and local guidance, in order to help practitioners and women make decisions about care [44]. Local variation from national guidance is perhaps inevitable, and may have benefits in terms of the ability to reflect local priorities and services, and promote increased local ownership and uptake [57]. It is also possible that in some services, in which MUs are less well established, local guidance may reasonably diverge from national guidance and evolve to converge with national guidance as the knowledge, skills and experience of MU midwives develops. We compared local guidance with national guidance in the form of NICE CG190 as an appropriate reference, not because all local guidance should 'conform' to this guidance. In some cases, variation from NICE may reflect the poor quality of underlying evidence, with some recommendations based on expert consensus. However, maternity services should be aware of the potential consequences of diverging from national guidance, and should as far as possible use robust methodology, refer to systematically identified evidence and consider equity. Small variations in recommendations can make a big difference to women who fall the 'wrong' side of the dividing line in admission guidelines [33]. Whilst half of the guidelines reviewed listed a referral pathway for women wishing to plan birth in the MU who fell outside of admission criteria, the specific limits proposed by local guidelines may affect women's own perception of their risk status [32] and so deter otherwise suitable women from considering birth in an MU on the grounds of safety. Large numbers of admission criteria may affect the practicalities of guideline implementation [58], influence professionals' perception of risk, and so impact on their birth place conversations with women [59]. Finally, the Birthplace study [9] provided much of the evidence about maternal and neonatal safety of planned birth in an MU for 'low risk' women, supporting the place of birth recommendations in NICE CG190. Birthplace used the guidance in the previous NICE intrapartum care guidelines [60] in order to classify women as 'low' or 'higher' risk. If local guidance about admission

criteria diverges substantially from national guidance, some women may be making birth place decisions based on an assumption of risk status that is not evidence-based, potentially resulting in unnecessary OU admission for some and elevated risks in MUs for others. It also means that in practice, many local admission guidelines may not be meeting the needs of women, or the midwives caring for them, who are working to navigate risk assessment in the context of all available birth place options [59,61]. Potential increases in the numbers of women who require individual assessment also means that changes in local structures and staffing may be required to ensure that women who need this receive the support of a senior midwife. Given the significant changes to the maternity care landscape since the Birthplace study [20,21], and the shift in MU admission criteria evidenced by our study, further research to investigate the safety of planned birth in different settings for specific groups of women, and a review of national guidance, may soon be required.

## Conclusions

This study found wide variation in local guideline layout and content, with frequent departure from national guidance and a lack of consensus regarding the parameters of straightforward pregnancy determining suitability for MU intrapartum care. This presents an inconsistent and potentially non-evidence based landscape for both women and the midwives responsible for facilitating women's decision-making about place of birth that may be deterring some women from choosing MUs and inadvertently excluding others. Where a lack of relevant evidence underlies the variation found, further research into outcomes for specific groups of women by planned place of birth is needed. The extent of variation from national guidance indicates that a review of that guidance may also be required in order to enable women to make birth place decisions with confidence.

## Supporting information

**S1 File. Criteria to be assessed for women planning place of birth, from NICE CG190.**
(DOCX)

**S2 File. Characteristics of responding and non-responding maternity services.**
(DOCX)

**S3 File. Individual admission criteria that were 'more inclusive' than NICE CG190.**
(DOCX)

**S4 File. Individual admission criteria that were 'more restrictive' than NICE CG190.**
(DOCX)

**S1 Table. Number of midwifery units of each type in each country of the UK at the time of the survey.**
(DOCX)

**S2 Table. Organisational alongside midwifery unit (AMU) admission policy (opt-in or opt-out) by characteristics of maternity service.**
(DOCX)

## Acknowledgments

The authors would like to thank: Paula Jenkins, UKMidSS Research Midwife, for her role in helping administer the survey; Anshita Shrivastava for help programming the online survey; Marian Knight and Julia Sanders for their comments on the draft manuscript; the UKMidSS

Steering Group for their advice about interpretation; and all the UKMidSS reporters and other midwives who took part in this study.

## Author Contributions

**Conceptualization:** Ceri Glenister, Ethel Burns, Rachel Rowe.

**Data curation:** Ceri Glenister.

**Formal analysis:** Ceri Glenister.

**Funding acquisition:** Rachel Rowe.

**Investigation:** Ceri Glenister, Ethel Burns.

**Methodology:** Ethel Burns, Rachel Rowe.

**Project administration:** Ceri Glenister.

**Supervision:** Ethel Burns, Rachel Rowe.

**Writing – original draft:** Ceri Glenister.

**Writing – review & editing:** Ceri Glenister, Ethel Burns, Rachel Rowe.

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
