## [Decision Letter · Decision Letter 0]

16 Apr 2020

PONE-D-20-02225

Local guidelines for admission to UK midwifery units compared with national guidance: a national survey using the UK Midwifery Study System (UKMidSS)

PLOS ONE

Dear Dr Rowe,

Thank you for submitting your manuscript to PLOS ONE. After careful consideration, we feel that it has merit but does not fully meet PLOS ONE’s publication criteria as it currently stands. Therefore, we invite you to submit a revised version of the manuscript that addresses the points raised during the review process.

We would appreciate receiving your revised manuscript by May 31 2020 11:59PM. To enhance the reproducibility of your results, we recommend that if applicable you deposit your laboratory protocols in protocols.io, where a protocol can be assigned its own identifier (DOI) such that it can be cited independently in the future. For instructions see: http://journals.plos.org/plosone/s/submission-guidelines#loc-laboratory-protocols

We look forward to receiving your revised manuscript.

Kind regards,

Karen Anne Grimmer, PhD

Academic Editor

PLOS ONE

Additional Editor Comments (if provided):

Well done on a nice paper. Please attend to the comments of the reviewer who would like a few changes made to the paper before it is accepted for publication.

Reviewers' comments:

Reviewer's Responses to Questions

**Comments to the Author**

1. Is the manuscript technically sound, and do the data support the conclusions?

Reviewer #1: Yes

Reviewer #2: Yes

2. Has the statistical analysis been performed appropriately and rigorously? 

Reviewer #1: Yes

Reviewer #2: Yes

3. Have the authors made all data underlying the findings in their manuscript fully available?

Reviewer #1: No

Reviewer #2: No

4. Is the manuscript presented in an intelligible fashion and written in standard English?

Reviewer #1: Yes

Reviewer #2: Yes

5. Review Comments to the Author

Reviewer #1: Dear authors

I have read your manuscript with interest. The survey is important to enable equal services and evidence based care for labouring women, although it is important to allow for some local variations to reflect local settings and resources. The method used is adequate to investigate the matter. I find the text well written and easy to understand. The pragmatic approach and easy to understand descriptive statistics are adequate to present differences in admission guidelines and represent real life practice. The results showed guidelines both with more inclusive as well as more restrictive criteria and it is interesting to see that 55-67 percent allowed for higher parity, age and BMI. The results might be useful in revision of local and national guidelines.

Figure 1 could include reasons for numbers of guidelines analysed

Reviewer #2: This paper presents information that will be of interest to midwives; particularly those working in or interested in working in MUs.

The paper tends to focus on England specific policy and MU provision. However, it is reporting on a national survey using the UKMidSS and should therefore provide information on the four countries of the UK. Specific examples of where this has not happened are included below but are apparent throughout the paper. This is particularly important as MUs are generally growing in number outside of England.

For example: References to support the assertion in the first sentence should include those from all of the 4 countries in the UK, not just England, for example, Better Start (2017); NI Maternity Strategy (2012). Also at the end of paragraph one, references should include those from all of the 4 countries in the UK, not just England. NHS relates to NHS Scotland, NHS Wales and England - in Northern Ireland, the health service is referred to as Health and Social Care (HSC) which reflects the integrated service provision of health and social care.

Consideration should be given to using using the term maternity care provider, rather than NHS organisations.

P.3, the numbers of MUs in England is detailed but not the rest of the UK; the number of MUs in all countries of the UK should be provided.

P.6 Data was extracted by CG; was the data extraction validated i.e. was it checked by anyone else and if not, why not?

It is stated 'We compared each criterion listed in each guideline...'; it should indicate which authors did that.

P.7 Under ‘More inclusive’ than NICE CG190: it states 'Criteria for which NICE CG190 recommends OU birth, but where the MU offered individual assessment for women planning admission'- but NICE advocates multidisciplinary discussion and individualised assessment for women if necessary.

P. 8 Under Analysis, need to remind the reader what the 'other sources' were.

P.10 You state that 'Almost a third of guidelines (30%) did not list the author(s) or a guideline development

group'. However, of those that did include authorship, it would have been interesting to know how the MU guidelines had been developed, for example were they co-produced with women, were they multidisciplinary or developed by midwives only.

P.11 Table 2, should make it clear that number of births in NHS organisation.

You state 'had a small number of variant criteria (1-5) and a small number of these were otherwise similar in layout', I would suggest replacing the second 'a small number' with 'not many'.

P.14- You state- 'Few guidelines (<8%) were fully aligned with national guidance'- however as stated in CG190 P.81-'Putting recommendations into practice can take time. How long may vary from guideline to guideline, and depends on how much change in practice or services is needed. Implementing change is most effective when aligned with local priorities'.

It is important to remember that local guidelines may be developed for where individual services are or where they want to get to and may align more fully with regional guidelines as services develop and MU midwives develop their experience, knowledge and skills. Where more than one MU uses the same guideline, it may require a guideline to be less or more restrictive to meet the needs of the evolutionary stage of each of the MUs using the guideline.

P.15- you state- 'This is the first UK-wide study to document AMU admission policy'-what about FMUs?

P.16- You state- 'none of the guidelines reviewed mentioned gestation in relation to late booking or fragmented antenatal care'.; does social services input cover this?

P.17 You state the need to be careful that guidelines 'do not disproportionately deny access to midwifery-led care for women from minority and socioeconomically deprived backgrounds- is social services input criteria and the inclusion of <16 years by some guidelines an attempt to address this to some degree?

P.18. You state -Whilst half of guidelines listed a referral pathway' it should be- Whilst half of the guidelines reviewed listed a referral pathway- please amend.

P.18 It is interesting that you state that women and midwives.. 'who are working to navigate risk in their discussions about birth place choices'.[54, 56]. It is important to consider that risk is often only raised when MU and Home are considered as place of birth. There are many iatrogenic risks of giving birth in an OU, but these are rarely discussed with women.

It would have been interesting to ask MU's whose guidance did not align with NICE guidance, why this was the case?

The supporting information files, provide useful additional data, although not all of te titles reflect the content of the tables, e.g. table 1 (S3). However, a table detailing the number of FMU's/AMUs in each of the four countries of the UK would be helpful.

There are some minor typos, please proof read again following amendments.

I hope this feedback is helpful.

6. PLOS authors have the option to publish the peer review history of their article (what does this mean?). If published, this will include your full peer review and any attached files.

Reviewer #1: No

Reviewer #2: No

---

## [Author Response · Author response to Decision Letter 0]

4 Jun 2020

Editor

Well done on a nice paper. Please attend to the comments of the reviewer who would like a few changes made to the paper before it is accepted for publication. 

Response

Thank you for your positive comment about our paper. We have responded to the reviewer’s comments below.

Reviewer #1 

I have read your manuscript with interest. The survey is important to enable equal services and evidence based care for labouring women, although it is important to allow for some local variations to reflect local settings and resources. The method used is adequate to investigate the matter. I find the text well written and easy to understand. The pragmatic approach and easy to understand descriptive statistics are adequate to present differences in admission guidelines and represent real life practice. The results showed guidelines both with more inclusive as well as more restrictive criteria and it is interesting to see that 55-67 percent allowed for higher parity, age and BMI. The results might be useful in revision of local and national guidelines.

Figure 1 could include reasons for numbers of guidelines analysed.

Response

Thank you for your extremely positive comments about our paper. 

With regard to Figure 1, we have revised this to make it clearer, including adding reasons for the numbers of guidelines analysed. 

Reviewer #2 

The paper tends to focus on England specific policy and MU provision. However, it is reporting on a national survey using the UKMidSS and should therefore provide information on the four countries of the UK. Specific examples of where this has not happened are included below but are apparent throughout the paper. This is particularly important as MUs are generally growing in number outside of England.

Response

Thank you for making this important point. Our intention was certainly to ensure that the focus of the paper was the whole of the UK, not just England, but we agree that we may not have given sufficient attention to this. This may be, in part, because much of the published evidence, about numbers of MUs etc, is based on English data, rather than data from the whole of the UK. We have made a number of changes to improve the paper in this regard, detailed below.

For example: References to support the assertion in the first sentence should include those from all of the 4 countries in the UK, not just England, for example, Better Start (2017); NI Maternity Strategy (2012). Also at the end of paragraph one, references should include those from all of the 4 countries in the UK, not just England. NHS relates to NHS Scotland, NHS Wales and England - in Northern Ireland, the health service is referred to as Health and Social Care (HSC) which reflects the integrated service provision of health and social care. 

Response

We have added references to key documents from all four countries of the UK in this paragraph and reworded some sentences to make it clear that we are referring to the whole of the UK. We have also made it clear that in Northern Ireland the health service is referred to as Health and Social Care.

Consideration should be given to using using the term maternity care provider, rather than NHS organisations.

Response

Thank you for this suggestion. We can see the rationale for this change, but are concerned that ‘maternity care provider’ could be interpreted as referring to individual people rather than service organisations. We have therefore used the term ‘maternity service’ instead.

P.3, the numbers of MUs in England is detailed but not the rest of the UK; the number of MUs in all countries of the UK should be provided. 

Response

This is largely because there is little published data relating to the rest of the UK. We have added some data about the number of MUs in other countries of the UK.

P.6 Data was extracted by CG; was the data extraction validated i.e. was it checked by anyone else and if not, why not?

Response

This was a student project carried out by CG for her Masters dissertation. Because of time and resource constraints it was not possible for duplicate data extraction to be carried out so CG did this alone with frequent discussion with RR, particularly when any uncertainties arose. We have added to our discussion of limitations to refer to this point.

It is stated 'We compared each criterion listed in each guideline...'; it should indicate which authors did that. 

Response

We have now revised the paper so it is clear that this was carried out by CG.

P.7 Under ‘More inclusive’ than NICE CG190: it states 'Criteria for which NICE CG190 recommends OU birth, but where the MU offered individual assessment for women planning admission'- but NICE advocates multidisciplinary discussion and individualised assessment for women if necessary. 

Response

We agree, NICE CG190 does advocate multidisciplinary discussion and individualised assessment for women if necessary. However, we set out to operationalise the guidance set out in the NICE CG190 chapter on planning place of birth and apply this to the guidelines analysed. For the criteria in Tables 6-7 in NICE CG190 it is advised that the woman plans birth in an obstetric unit. For the criteria listed in Tables 8-9 in NICE CG190, individualised assessment is recommended in considering planned place of birth. When it was clear, in a guideline we analysed, that women with a condition or factor listed in Tables 6-7 could be considered eligible for admission to the midwifery unit following individualised assessment, we considered this to be ‘more inclusive’ than NICE CG190.

P. 8 Under Analysis, need to remind the reader what the 'other sources' were. 

Response

Thank you, this is helpful. We have added a reminder about what the ‘other sources’ were.

P.10 You state that 'Almost a third of guidelines (30%) did not list the author(s) or a guideline development

group'. However, of those that did include authorship, it would have been interesting to know how the MU guidelines had been developed, for example were they co-produced with women, were they multidisciplinary or developed by midwives only. 

Response

Thank you for raising this, it is a really good point. Unfortunately we did not extract any data from the guidelines about the composition of the authorship or development group. However, we did extract data from the guidelines in relation to whether there was any evidence of the involvement of service users in the development of the guideline. We have added this information to Table 1 on page 12 and at lines 217-9 on page 11. In making these changes to Table 1 we noticed some errors, arising from copying by mistake from an earlier version. These have now been corrected. We have also checked other tables. 

P.11 Table 2, should make it clear that number of births in NHS organisation. 

Response

Thank you, we have added a footnote to clarify this.

You state 'had a small number of variant criteria (1-5) and a small number of these were otherwise similar in layout', I would suggest replacing the second 'a small number' with 'not many'. 

Response

Thank you for pointing this out, we have changed the wording for this sentence.

P.14- You state- 'Few guidelines (<8%) were fully aligned with national guidance'- however as stated in CG190 P.81-'Putting recommendations into practice can take time. How long may vary from guideline to guideline, and depends on how much change in practice or services is needed. Implementing change is most effective when aligned with local priorities'. 

Response

We completely agree – this is an important point and we have discussed this issue in detail on page 18. The statement on page 14 is simply a factual summary of the key findings so we don’t think it would be appropriate to add to it. 

It is important to remember that local guidelines may be developed for where individual services are or where they want to get to and may align more fully with regional guidelines as services develop and MU midwives develop their experience, knowledge and skills. Where more than one MU uses the same guideline, it may require a guideline to be less or more restrictive to meet the needs of the evolutionary stage of each of the MUs using the guideline. 

Response

Thank you for making this point. We have discussed related issues on page 18, but have added a sentence reflecting the reviewer’s comment.

P.15- you state- 'This is the first UK-wide study to document AMU admission policy'-what about FMUs? 

Response

We use the phrase “admission policy” to refer very specifically to AMUs and whether they are regarded as the ‘default option’ for eligible women (an ‘opt-out’ model) or whether women have to actively choose to plan birth there (an ‘opt-in’ model). This is not an approach taken for FMUs in the UK. We defined this at the end of the background section on page 5, under data and sources on the same page and in the results section on page 10. We have added to the sentence referred to on page 15 to clarify.

P.16- You state- 'none of the guidelines reviewed mentioned gestation in relation to late booking or fragmented antenatal care'.; does social services input cover this? 

Response

We’re not sure that we completely understand the reviewer’s point here. The sentence referred to on page 16 relates specifically to admission criteria which exclude women who book late for care, or have fragmented antenatal care, from access to midwifery units. One possible interpretation of these apparently ‘more restrictive’ criteria is that they are a way to ensure that women for whom gestational age is uncertain or outside the usual criteria for admission to midwifery-led care, are not admitted to a midwifery unit. However, as this sentence explains, we found no mention of gestation in any of the guidelines reviewed in relation to late booking or fragmented antenatal care. As shown in the supplementary S4 file, some guidelines also included criteria which excluded women with social services input from admission to the midwifery unit, but we’re not sure how this relates to the reviewer’s comment. Social services input would not be a given for women who book late or choose not to access antenatal care in the UK.

P.17 You state the need to be careful that guidelines 'do not disproportionately deny access to midwifery-led care for women from minority and socioeconomically deprived backgrounds- is social services input criteria and the inclusion of <16 years by some guidelines an attempt to address this to some degree? 

Response

NICE CG190 guidance does not consider age<16 years or social services input as reasons to advise women not to plan birth in a midwifery unit. The only way in which these criteria were used in guidelines were as ‘more restrictive’ criteria which could potentially exclude women from the midwifery unit. For example, this included guidelines where women with social services input or aged <16 years were admitted to the midwifery unit only after individual assessment or were explicitly excluded from admission. These are shown in the tables in supplementary file S4. 

P.18. You state -Whilst half of guidelines listed a referral pathway' it should be- Whilst half of the guidelines reviewed listed a referral pathway- please amend. 

Response

Thank you for pointing this out – we have amended this.

P.18 It is interesting that you state that women and midwives.. 'who are working to navigate risk in their discussions about birth place choices'.[54, 56]. It is important to consider that risk is often only raised when MU and Home are considered as place of birth. There are many iatrogenic risks of giving birth in an OU, but these are rarely discussed with women. 

Response

We completely agree with your point, but feel that it would be beyond the scope of this paper to bring this into our discussion. We have reworded this sentence very slightly so that it now reads “who are working to navigate risk assessment in the context of all available birth place options”.

It would have been interesting to ask MU's whose guidance did not align with NICE guidance, why this was the case?

Response

We agree, this would have been really interesting. However, at the point when we carried out our survey and collected guidelines we did not know which service providers had guidance that did not align with NICE, and we did not have the time or resources to go back to participants afterwards. We have added a sentence to the strengths and limitations section acknowledging this as a potential limitation.

The supporting information files, provide useful additional data, although not all of te titles reflect the content of the tables, e.g. table 1 (S3). However, a table detailing the number of FMU's/AMUs in each of the four countries of the UK would be helpful. 

Response

We have checked all supporting information files and revised where necessary. We don’t recognise that the title of Table 1 in S3 File does not reflect the content. We have added to the title, and to those of some other supplementary tables, to help clarify what is shown. We have added a new S1 Table detailing the number of midwifery units of each type in each of the four countries of the UK at the time the survey was carried out.

There are some minor typos, please proof read again following amendments.

I hope this feedback is helpful. 

Response

Thank you, we have been over the whole manuscript and supplementary tables again and corrected any typos we found. 

We are extremely grateful to you for reviewing our paper so carefully and thoughtfully. Your comments have resulted in changes which have improved the paper.

---

## [Decision Letter · Decision Letter 1]

4 Sep 2020

Local guidelines for admission to UK midwifery units compared with national guidance: a national survey using the UK Midwifery Study System (UKMidSS)

PONE-D-20-02225R1

Dear Dr. Rachel Rowe,

We’re pleased to inform you that your manuscript has been judged scientifically suitable for publication and will be formally accepted for publication once it meets all outstanding technical requirements.

Kind regards,

Sharon Mary Brownie

Academic Editor

PLOS ONE

Reviewers' comments:

Reviewer's Responses to Questions

**Comments to the Author**

1. If the authors have adequately addressed your comments raised in a previous round of review and you feel that this manuscript is now acceptable for publication, you may indicate that here to bypass the “Comments to the Author” section, enter your conflict of interest statement in the “Confidential to Editor” section, and submit your "Accept" recommendation.

Reviewer #1: All comments have been addressed

Reviewer #2: All comments have been addressed

2. Is the manuscript technically sound, and do the data support the conclusions?

Reviewer #1: Yes

Reviewer #2: Yes

3. Has the statistical analysis been performed appropriately and rigorously? 

Reviewer #1: Yes

Reviewer #2: Yes

4. Have the authors made all data underlying the findings in their manuscript fully available?

Reviewer #1: No

Reviewer #2: Yes

5. Is the manuscript presented in an intelligible fashion and written in standard English?

Reviewer #1: Yes

Reviewer #2: Yes

6. Review Comments to the Author

Reviewer #1: Dear authors

I find this paper well written and interesting and had very few comments to the text in the first round. I do see that you now have taken my comments into consideration as well as the comments from reviewer #2.

Reviewer #2: My sincere apologies for ny delay in returning my review but this was unavoidable due to other workload and leave. I feel that the authors have responded positively and fully to my comments.

Please add HSC Trusts to the statements that refer to NHS Trusts and Health Boards

7. PLOS authors have the option to publish the peer review history of their article (what does this mean?). If published, this will include your full peer review and any attached files.

Reviewer #1: No

Reviewer #2: No

---

## [Editor Report · Acceptance letter]

8 Sep 2020

PONE-D-20-02225R1 

Local guidelines for admission to UK midwifery units compared with national guidance: a national survey using the UK Midwifery Study System (UKMidSS) 

Dear Dr. Rowe:

I'm pleased to inform you that your manuscript has been deemed suitable for publication in PLOS ONE. Congratulations! Your manuscript is now with our production department. 

Kind regards, 

on behalf of

Professor Sharon Mary Brownie 

Academic Editor

PLOS ONE